# Mitigating Input Noise in Binary Classification: A Unified Framework with Data Augmentation

## Abstract

Classification techniques have achieved significant success across fields such as computer vision, information retrieval, and natural language processing. However, much of this progress assumes input features are error-free – a condition rarely met in practice. In real-world scenarios, noisy inputs caused by measurement errors are common, leading to biased or suboptimal classification results. This paper presents a unified framework for binary classification with noisy inputs, offering a generalizable solution that applies across various supervised learning algorithms and noise models. We provide a theoretical analysis of the bias introduced by ignoring input noise (also referred to as feature corruption) and identify conditions where this bias can be safely disregarded. To address cases where noise correction is needed, we propose a novel data augmentation-based method to mitigate input noise effects. Our approach is both comprehensive and theoretically grounded, providing practical solutions for improving classification accuracy in noisy data enviroments. Extensive experiments, including analyses of medical image datasets, demonstrate the superior performance of our methods under different noise conditions.

## 1 Introduction

With the exponential growth of data across diverse domains, classification techniques have emerged as indispensable tools in solving complex problems in fields, such as computer vision (Krizhevsky, Sutskever, and Hinton 2012), information retrieval (Pang et al. 2017), and natural language processing (Howard and Ruder 2018). Widely used methods like logistic regression, support vector machines, boosting, and neural networks have demonstrated remarkable success in numerous applications (Mohri, Rostamizadeh, and Talwalkar 2018), but much of their effectiveness hinges on the assumption that the input data are clean and error-free. In practice, this assumption rarely holds. Noisy, imprecise, or corrupted features are prevalent in practice, creating substantial challenges for these models and leading to suboptimal or biased results.

One of the key challenges in supervised learning is dealing with noisy input features. While research on handling noisy labels has been prolific – spanning data-cleaning techniques, robust loss functions, and probabilistic methods (e.g., Song et al. 2022), there has been comparatively less attention given to noisy inputs (or corrupted features), where measurement errors affect feature values. Addressing noisy inputs remains an interesting research area, particularly as real-world data collection processes are rarely flawless.

### 1.1 Relative Work

Research on classification with noisy inputs spans both traditional machine learning algorithms like discriminative methods (e.g., Fidler, Skocaj, and Leonardis 2006; Adeli et al. 2018), logistic regression (e.g., Stefanski and Carroll 1985) and support vector machines (Rabaoui et al. 2008), as well as deep learning techniques, particularly in the context of noisy images. In deep learning, existing methods can be broadly

classified into two categories. The first category involves preprocessing approaches, where denoising techniques are applied to reconstruct clean images from images from noisy ones before passing them through a convolutional neural network (CNN) for classification (e.g., Roy, Ahmed, and Akhand 2018). However, the success of this approach heavily depends on the quality of the denoising step, which introduces additional complexity and uncertainty. A useful example is FROM (Face Recognition with Occlusion Masks) by Qiu et al. (2021), which detects corrupted features using a CNN and dynamically cleans them with learned masks. In the second approach, instead of attempting to recover the clean images, a noise-robust CNN architecture is designed to directly classify the noisy images (e.g., Momeny et al. 2021). This direct approach reduces the dependency on preprocessing and has shown promise in noisy image classification tasks.

Beyond deep learning, non-parametric approaches such as Gaussian processes (e.g., Seeger 2002; Kuss, Rasmussen, and Herbrich 2005; Nickisch and Rasmussen 2008; Hernández-lobato, Hernández-lobato, and Dupont 2011; Rodrigues, Pereira, and Ribeiro 2014; Zhao et al. 2021) have garnered increased attention for handling noisy inputs in multi-class classification problems (Villacampa-Calvo et al. 2021). For example, Hernández-lobato et al. (2014) presented a Gaussian process classification method that treats privileged information as noise.

Another emerging area is fair classification (e.g., Donini et al. 2018; Huang and Vishnoi 2019; Zafar et al. 2019; Agarwal et al. 2018; Hardt, Price, and Srebro 2016) under noisy conditions, particularly when protected attributes are noisy. Lamy et al. (2019) demonstrated that fairness can still be achieved in classifiers with noisy binary protected attributes, provided specific fairness measures such as the mean-difference score are used. Celis et al. (2021) extended this work to the non-binary case, developing optimization frameworks that enable fair classification even when the protected attributes are noisy.

In addition, robust machine learning has been approached by deliberately corrupting features to train models (e.g., Burges and Schölkopf 1996; Globerson and Roweis 2006; Dekel and Shamir 2008; Xu, Caramanis, and Mannor 2009). For example, Bahri et al. (2022) proposed SCARF, a technique for contrastive learning that involves corrupting random subsets of features; Maaten et al. (2013) introduced a robust learning method that corrupts features using noise sampled from known distributions and minimizes the expected loss under the corrupting distribution.

Despite these advances, several limitations persist in the current literature. Most existing methods are tailored to specific algorithms and often rely on simple input noise models (e.g., additive noise), commonly referred to as measurement error models in the statistical literature (Yi 2017; Yi, Delaigle, and Gustafson 2021). Moreover, few studies offer a theoretical framework for analyzing the impact of input noise on classifier performance or propose generalizable correction methods that can be applied across a broad range of classification problems. This leaves open questions about how to effectively handle input nose across different machine learning algorithms and noise structures, especially in large-scale and high-dimensional settings.

## 1.2 OUR CONTRIBUTIONS

In this paper, we take a significant step toward closing these gaps by focusing on binary classification with noisy inputs. We present a unified framework for addressing noisy inputs, with theoretical guarantees and a practical correction method that applies across a wide range of classification algorithms. Our key contributions are as follows:

- A General Classification Framework with Noisy Inputs: We develop a general framework for binary classification that explicitly accounts for input noise. Using the commonly employed 0-1 loss to evaluate classifiers, we address the computational challenges by utilizing a convex surrogate loss function, denoted as $\varphi(\cdot)$, and use the corresponding $\varphi$-risk as a metric to evaluate classifier

performance. This approach provides a flexible, robust framework that extends beyond specific algorithms or noise models.

- Theoretical Analysis of Input Noise: We provide a rigorous theoretical analysis of the bias introduced by the naive approach that ignores input noise. Our analysis yields an informative upper bound for the disparity between the generalization error and $\varphi$-risk of the optimal classifier obtained from the naive procedure. Notably, this upper bound shrinks as the noise level decreases, identifying cases where ignoring input noise can be safely disregarded. This result is critical, as it offers insight into when and how noise affects classifier performance and provides guidance on the conditions under which noise correction is necessary.

- A Novel Correction Method via Data Augmentation: To mitigate the effect of noisy inputs, we propose a novel correction method by augmenting the dataset with newly generated data that either are precisely measured or contain minimal error. This augmented dataset enables us to devise robust classifiers that mitigate the bias induced by noisy inputs. Unlike previous methods, our approach is model-agnostic and can be applied to a broad class of classification algorithms, making it highly versatile for different applications.

- Extensive Empirical Evaluation: We validate the proposed method through extensive numerical experiments. We first apply our correction method to a real-world chest X-ray image dataset to demonstrate its effectiveness in a practical healthcare setting. We then conduct a series of synthetic experiments to assess the performance of our method under different noise levels and input distributions. The results consistently demonstrate the superior performance of the proposed method.

By providing a unified and generalizable approach to handling noisy inputs, this paper makes important contributions to the field of classification, addressing noisy inputs in a comprehensive and theoretically grounded manner.

The rest of this paper is structured as follows: In Section 2, we introduce the general classification framework with accurately measured inputs. Section 3 extends this framework to noisy inputs and presents our correction method. We evaluate our approach in Section 4, using both real-world and synthetic datasets to assess its effectiveness.

## 2 CLASSIFICATION FRAMEWORK

Let $\mathcal{X} \subseteq \mathbb{R}^p$ denote the input space, equipped with the Borel $\sigma$-algebra $\sigma_{\mathcal{X}}$, where $p$ is the number of features, and let $\mathcal{Y} = \{-1, +1\}$ denote the output (or label) space, endowed with the $\sigma$-algebra $\sigma_{\mathcal{Y}}$. Let $X$ denote the $p$-dimensional input vector taking values in $\mathcal{X}$, and let $Y$ represent the binary output variable taking values in $\mathcal{Y}$. Let $\mathcal{D}$ denote the joint distribution of $X$ and $Y$. Let $\mathcal{H}$ denote the set of all measurable functions from the input measurable space $(\mathcal{X}, \sigma_{\mathcal{X}})$ to the output measurable space $(\mathcal{Y}, \sigma_{\mathcal{Y}})$. For any $h \in \mathcal{H}$, the *generalization error*, or risk is defined as:

$$R(h) \triangleq \mathbb{E}\{\mathbb{1}_{\{h(X) \neq Y\}}\} \tag{1}$$

where the expectation is taken with respect to the joint distribution $\mathcal{D}$ of $X$ and $Y$, and $\mathbb{1}_{\{h(X) \neq Y\}}$ is the *0-1 loss* function, equal to 1 if the classifier $h$ misclassifies the label of $X$ and 0 otherwise.

Our goal is to find a classifier $h_0 \in \mathcal{H}$ that minimizes the *generalization error* :

$$h_0 \in \arg \min_{h \in \mathcal{H}} R(h), \tag{2}$$

where the symbol "$\in$" indicates that the solutions of (2) may not be unique.

A well-known solution to (2) is the *Bayes classifier*, given by $h_0(X) = sign\left\{\eta(X) - \frac{1}{2}\right\}$ (Boucheron, Bousquet, and Lugosi 2005, Section 2), where $\eta(X) \triangleq \mathbb{P}(Y = 1|X)$, and the sign function $sign(t)$ is defined as 1 if $t \geq 0$ and $-1$ if $t < 0$.

Let $R_0 \triangleq R(h_0)$ represent the *generalization error* of the *Bayes classifier*. Then, we have

$$R_0 = \min_{h \in \mathcal{H}} R(h). \tag{3}$$

While the *Bayes classifier* is conceptually optimal, it often proves impractical because we typically do not know the distribution $\mathcal{D}$ of $X$ and $Y$. To address this challenge, we can focus on a subset of $\mathcal{H}$ instead of the entire space.

Typically, we select a subset of $\mathcal{H}$ that possesses favorable mathematical properties, such as a set of bounded linear functions that includes the *Bayes classifier*. A common approach to construct such a subset is to specify a family of measurable functions from $\mathcal{X}$ to $\mathbb{R}$, denoted $\mathcal{F}$, and define our subset of $\mathcal{H}$ as $sign(\mathcal{F}) \triangleq \{sign \circ f : f \in \mathcal{F}\}$, where $sign \circ f$ represents the composition of functions $sign$ and $f$. Consequently, we seek to find the minimizer $\arg\min_{f \in \mathcal{F}} R(sign \circ f)$ for an appropriately chosen class $\mathcal{F}$.

Often, we consider a class $\mathcal{F}$ with *Rademacher complexity* $\mathcal{R}(\mathcal{F})$ on the order of $\mathcal{O}(n^{-\frac{1}{2}})$ (Boucheron, Bousquet, and Lugosi 2005). Further, we assume that the functions in $\mathcal{F}$ are uniformly bounded, meaning there exists a constant $C_{\mathcal{F}} > 0$ such that $|f(x)| \leq C_{\mathcal{F}}$ for all $f \in \mathcal{F}$ and $x \in \mathcal{X}$. Since $sign(f) = sign(\frac{f}{C_{\mathcal{F}}})$ for any $f \in \mathcal{F}$, we can equivalently consider the set $\mathcal{F}' \triangleq \{\frac{f}{C_{\mathcal{F}}} : f \in \mathcal{F}\}$. Thus, without loss of generality, we assume that $|f(x)| \leq 1$ for all $f \in \mathcal{F}$ and $x \in \mathcal{X}$.

To simplify our discussion, we will refer to any function $f \in \mathcal{F}$ as a classifier throughout the paper while keeping in mind that it is actually $sign(f(x))$ that predicts the label for $x$. Define the loss function as

$$\ell(u) = \mathbb{1}_{\{u \in [0, \infty)\}} \quad \text{for any } u \in \mathbb{R}. \tag{4}$$

Using (1) and (4), we express the *generalization error* of the classifier $sign \circ f$ as:

$$R(sign \circ f) = \mathbb{E}\left\{\mathbb{1}_{\{sign \circ f(X) \neq Y\}}\right\} = \mathbb{E}\{\ell(-Yf(X))\},$$

which we will denote simply as

$$R(f) \triangleq \mathbb{E}\{\ell(-Yf(X))\}. \tag{5}$$

We thereby aim to find a classifier from $\mathcal{F}$ that minimizes the *generalization error* $R(f)$. However, the non-convexity of $\ell(u)$ complicates this minimization. As a remedy, we consider a *convex surrogate* function

$$\varphi : \mathbb{R} \to \mathbb{R}^+ \tag{6}$$

that serves as an upper bound for the loss function $\ell(u)$ and is Lipschitz continuous restricted on the interval $[-1, 1]$. Specifically, we require that

(a). $\ell(u) \leq \varphi(u)$ for all $u \in \mathbb{R}$;

(b). there exists a positive constant $L_{\varphi}$ such that

$$|\varphi(u_1) - \varphi(u_2)| \leq L_{\varphi}|u_1 - u_2| \quad \text{for all } u_1, u_2 \in [-1, 1]. \tag{7}$$

Surrogate functions that meet these criteria are bounded, as shown in the following lemma, whose proof is deferred to Appendix B.1.

**Lemma 1.** *For any convex surrogate $\varphi(\cdot)$ defined in (6), there exists a constant $E_\varphi > 0$ such that*

$$|\varphi(f(x))| \leq E_\varphi \quad \text{for all } f \in \mathcal{F} \text{ and } x \in \mathcal{X}. \tag{8}$$

By substituting $\ell(\cdot)$ in (5) with a convex surrogate $\varphi(\cdot)$, we define the *$\varphi$-risk* for $f \in \mathcal{F}$ (e.g., Lopez-Paz et al. 2015) as:

$$R_\varphi(f) \triangleq \mathbb{E}\{\varphi(-Yf(X))\}. \tag{9}$$

Now, given a convex surrogate $\varphi$, as defined in (6), our objective is to find the optimal classifier $f_0 \in \mathcal{F}$, determined by

$$f_0 = \arg\min_{f \in \mathcal{F}} R_\varphi(f), \tag{10}$$

which can be readily found using convex optimization algorithms due to the convexity of $\varphi$.

The $\varphi$-risk provides a mathematically convenient upper bound for the original *risk* in (5). Although different surrogate functions can yield varying upper bounds, a well-calibrated surrogate function $\varphi(\cdot)$ can closely approximate $\ell(\cdot)$ and facilitate meaningful risk bounds. To this end, Bartlett et al. (2006) introduced a useful class of surrogate functions known as *classification-calibrated* convex surrogates, which include commonly used losses such as the logistic loss $\varphi(u) = \log_2(1 + \exp(u))$ used in logistic regression, the *hinge loss* $\varphi(u) = \max\{0, 1 + u\}$ used in the *support vector machine (SVM)*, and the *exponential loss* $\varphi(u) = \exp(u)$ used in *Adaboost*.

While utilizing a convex surrogate function simplifies our minimization in (5) to a convex optimization problem, the unknown distribution $\mathcal{D}$ still prevents us from obtaining $f_0$ directly from (10), or even from $\arg\min_{f \in \mathcal{F}} R_\varphi(f)$. To address this issue, we consider a collection of $n$ independently and identically distributed (i.i.d.) copies of $\{X, Y\}$, denoted $S(n) = \{\{X_1, Y_1\}, \cdots, \{X_n, Y_n\}\}$, where $n$ is the sample size.

With the sample $S(n)$ available, we replace $R_\varphi(f)$ in (9) with the empirical $\varphi$-risk:

$$\hat{R}_\varphi(f) \triangleq \frac{1}{n} \sum_{i=1}^{n} \varphi(-Y_i f(X_i)) \tag{11}$$

and then find the empirical classifier $\hat{f}_\varphi \in \mathcal{F}$ by minimizing the empirical $\varphi$-risk:

$$\hat{f}_\varphi = \arg\min_{f \in \mathcal{F}} \hat{R}_\varphi(f). \tag{12}$$

To evaluate the performance of $\hat{f}_\varphi$, we compare it with the theoretical minimizer $h_0$ in (2) and the minimizer $f_0$ in (10) by examining the differences $R(\hat{f}_\varphi) - R_0$ and $R_\varphi(\hat{f}_\varphi) - R_\varphi(f_0)$, where $R_0$ is defined in (3). The first measure, $R(\hat{f}_\varphi) - R_0$, measures the difference in expected misclassification rates between the optimal classifier over $\mathcal{H}$ and our empirical classifier $\hat{f}_\varphi$. The second term compares the $\varphi$-risk of the empirical classifier with that of the optimal classifier over $\mathcal{F}$. While deriving explicit expressions for these differences is challenging, literature often focuses on identifying meaningful upper bounds for $R(\hat{f}_\varphi) - R_0$ and $R_\varphi(\hat{f}_\varphi) - R_\varphi(f_0)$, typically within the context of finite-dimensional input spaces (Boucheron, Bousquet, and Lugosi 2005; Mohri, Rostamizadeh, and Talwalkar 2018).

To extend these concepts to broader applications, we develop our analysis in the context of infinite-dimensional input spaces and provide upper bounds of $R(\hat{f}_\varphi) - R_0$ and $R_\varphi(\hat{f}_\varphi) - R_\varphi(f_0)$. Contrast to $\mathcal{H}$ that is a superset of $sign(\mathcal{F})$ containing all measurable functions mapping from the input measurable space $(\mathcal{X}, \sigma_\mathcal{X})$ to the output measurable space $(\mathcal{Y}, \sigma_\mathcal{Y})$, we consider a superset of $\mathcal{F}$, denoted $\mathcal{G}$, which includes all measurable functions mapping from the input measurable space $(\mathcal{X}, \sigma_\mathcal{X})$ to the measurable space

$(\mathbb{R}, \mathcal{B}(\mathbb{R}))$, where $\mathcal{B}(\mathbb{R})$ is the Borel $\sigma$-algebra on $\mathbb{R}$. The introduction of $\mathcal{G}$ enables us to express $\mathcal{F}$ relative to $\mathcal{G}$ in the same manner as describing $sign(\mathcal{F})$ relative to $\mathcal{H}$.

**Theorem 1.** *Consider $\varphi(\cdot)$ defined in (6) with the constant $E_\varphi$ described in Lemma 1. Let $\delta$ be any constant between $0$ and $1$. Define*

$$C(n, \delta) = 4L_\varphi \mathcal{R}(\mathcal{F}) + 2E_\varphi \sqrt{\frac{2log(1/\delta)}{n}} \tag{13}$$

*and*

$$D_\varphi(n, \delta) = R_\varphi(f_0) - \min_{g \in \mathcal{G}} R_\varphi(g) + C(n, \delta). \tag{14}$$

*where $f_0$ is given by (10). Then, with probability at least $1 - \delta$,*

$$R_\varphi(\hat{f}_\varphi) - R_\varphi(f_0) \le C(n, \delta). \tag{15}$$

*Furthermore, if $\varphi$ is classification-calibrated, there exists a nondecreasing continuous function $\zeta_\varphi : \mathbb{R} \to [0, 1]$, with $\zeta_\varphi(0) = 0$, such that with probability at least $1 - \delta$,*

$$R(\hat{f}_\varphi) - R_0 \le \zeta_\varphi(D_\varphi(n, \delta)). \tag{16}$$

The proof of Theorem 1 is presented in Appendix B.2. This theorem is applicable to a broad class of convex surrogates $\varphi(\cdot)$, including the *hinge loss*, the *exponential loss*, and the *logistic loss*, all of which are Lipschitz continuous. Setting $\delta = \frac{1}{n}$, the upper bound (15) indicates that $R_\varphi(\hat{f}_\varphi)$ converges to the true $\varphi$-risk $R_\varphi(f_0)$ in probability as the sample size $n$ approaches infinity, implying that $\hat{f}_\varphi$ is $\varphi$-*consistent* (Definition A.2 in the appendix). Furthermore, if the class $\mathcal{F}$ includes the minimum of $R_\varphi(g)$, the upper bound in (16) shows that expected misclassification rate $R(\hat{f}_\varphi)$ also converges to the optimal risk $R_0$ in probability as the sample size $n$ approaches infinity. This is because $\zeta_\varphi(\cdot)$ is continuous, and hence, $\hat{f}_\varphi$ derived from the empirical $\varphi$-risk is *consistent* (Definition A.1 in the appendix).

## 3 CLASSIFICATION WITH NOISY INPUTS

Theorem 1 provides guidelines for selecting an appropriate $\varphi$-function, valid only when the input variables $\{X_i : i = 1, \cdots, n\}$ are precisely measured. This condition is, however, often violated in practice, where mismeasurement of $X_i$ is common. We denote the observed version of $X_i$ as $X_i^*$ and assume access only to the sample $S^*(n) \triangleq \left\{ \{X_i^*, Y_i\} : i = 1, \cdots, n \right\}$, where the $X_i^*$ are assumed to be independent and may have different distributions for $i = 1, \cdots, n$.

For each $i = 1, \cdots, n$, we define the noise level as:

$$D_i \triangleq \mathbb{E}\{||X_i^* - X_i||_2^2\}, \tag{17}$$

where $||a||_2 \triangleq \sqrt{a^\mathsf{T} a}$ is the $L_2$-norm for vector $a$. To see how to determine $D_i$, we examine widely-used models in Appendix C.

Next, we study the impact of noisy inputs. With only surrogate measurements $X_i^*$ for $X_i$, it might be tempting to train a classifier by simply replacing $X_i$ with $X_i^*$, leading to what we call a *naive classifier*. In this context, we derive the *naive empirical $\varphi$-risk* and *naive classifier* by replacing $X_i$ with $X_i^*$ in (11) and (12), respectively:

$$\hat{R}_\varphi^*(f) \triangleq \frac{1}{n} \sum_{i=1}^n \varphi(-Y_i f(X_i^*)) \quad \text{and} \quad \hat{f}_\varphi^* = \arg \min_{f \in \mathcal{F}} \hat{R}_\varphi^*(f). \tag{18}$$

Similar to the discussion about (A.1) in the appendix, $\hat{f}_\varphi^*$ implicitly depends on the noisy sample $S^*(n)$, and we define

$$R(\hat{f}_\varphi^*) = \mathbb{E}\{\ell(-Y\hat{f}_\varphi^*(X))\big|S^*(n)\} \quad \text{and} \quad R_\varphi(\hat{f}_\varphi^*) = \mathbb{E}\{\varphi(-Y\hat{f}_\varphi^*(X))\big|S^*(n)\}$$

to capture the associated randomness.

For many settings different from the current context, it has been well documented that naive methods ignoring the feature of mismeasurement commonly yield biased results, with induced bias varies from problem to problem (e.g., Yi 2017). Here, we investigate the performance of the naive classifier $\hat{f}_\varphi^*$ in terms of a $\varphi$-*risk* and the *risk*. In Appendix B.3, we prove the following theorem which provides upper bounds for $\mathbb{E}\{R_\varphi(\hat{f}_\varphi^*) - R_\varphi(f_0)\}$ and $\mathbb{E}\{R(\hat{f}_\varphi^*) - R_0\}$.

**Theorem 2.** *Consider $\varphi(\cdot)$ in (6) with the constant $E_\varphi$ described in Lemma 1. Assume that all functions in $\mathcal{F}$ are Lipschitz continuous with respect to the $L_2$-norm in $\mathcal{X}$, with a common Lipschitz constant $L_\mathcal{F}$. That is, for any $f \in \mathcal{F}$ and $x, x' \in \mathcal{X}$,*

$$|f(x) - f(x')| \leq L_\mathcal{F}||x - x'||_2. \tag{19}$$

*Then*

$$\mathbb{E}\{R_\varphi(\hat{f}_\varphi^*) - R_\varphi(f_0)\} \leq C\Big(n, \frac{1}{n}\Big) + \frac{4E_\varphi}{n} + \frac{2L_\varphi L_\mathcal{F}}{n}\sum_{i=1}^n \sqrt{D_i}, \tag{20}$$

*where $C(\cdot, \cdot)$ is defined in (13). Furthermore, if $\varphi$ is classification-calibrated, then*

$$\mathbb{E}\{R(\hat{f}_\varphi^*) - R_0\} \leq \zeta_\varphi\left(D_\varphi\Big(n, \frac{1}{n}\Big) + \frac{4E_\varphi}{n} + \frac{2L_\varphi L_\mathcal{F}}{n}\sum_{i=1}^n \sqrt{D_i}\right), \tag{21}$$

*where $D_\varphi(\cdot, \cdot)$ is defined in (14), and $\zeta_\varphi(\cdot)$ is a nondecreasing function with $\zeta_\varphi(0) = 0$ as in Theorem 1.*

The upper bound (20) in Theorem 2 conveys an important message. By the limit property that $\lim_{n\to\infty} \frac{1}{n}\sum_{i=1}^n x_i = 0$ if $\lim_{n\to\infty} x_n = 0$ (Choudary and Niculescu 2014, Section 2.7), we find that $\lim_{n\to\infty} \frac{2L_\varphi L_\mathcal{F}}{n}\sum_{i=1}^n \sqrt{D_i} = 0$ if $D_n \to 0$ as $n \to \infty$. Further, under the assumption $\mathcal{R}(\mathcal{F}) = \mathcal{O}(\frac{1}{\sqrt{n}})$ in Section 2, we have $\lim_{n\to\infty}\left\{C\Big(n, \frac{1}{n}\Big) + \frac{4E_\varphi}{n}\right\} = 0$. Therefore, when $D_n \to 0$, we conclude that $\mathbb{E}\{R_\varphi(\hat{f}_\varphi^*) - R_\varphi(f_0)\}$ approaches zero. If the class $\mathcal{F}$ includes $\arg\min_{g\in\mathcal{G}} R_\varphi(g)$, then $R_\varphi(f_0) = \min_{f\in\mathcal{F}} R_\varphi(f)$, showing that if $D_\varphi(n, \frac{1}{n})$ in (14) converges to zero as $n \to \infty$, $\mathbb{E}\{R(\hat{f}_\varphi^*) - R_0\}$ converges to zero as $n \to \infty$. Consequently, as the input noise degree $D_n$ approaches zero as $n \to \infty$, the naive classifier $\hat{f}_\varphi^*$ is both $\varphi$-*consistent* and *consistent* (Definition A.2 and Definition A.1 in the appendix), showing that in this case, the input noise is ignorable asymptotically.

Building on the results in Theorem 2, we propose a correction method that construct an augmented dataset, combining the original noisy inputs and newly added data that either are precisely measured or contain minor error. Implementation details are provided in Algorithm 1. According to Theorem 2, if the size of the augmented dataset $\tilde{n}$ is sufficiently large, the classifier provided by Algorithm 1 can yield reliable learning outcomes.

## 4 EXPERIMENTS

### 4.1 SENSITIVITY ANALYSES OF MEDICAL IMAGE DATA

Chest X-rays are one of the most common imaging tests, crucial for screening, diagnoising and managing of various life-threatening diseases. CheXpert is a large chest radiography dataset that includes 224,316

---

**Algorithm 1** Implementation of Correction Method

---

1: **Input:** Start with the observed noisy data: $\mathcal{D}^* = \left\{ \{X_i^*, Y_i\} : i = 1, \cdots, n \right\}$
2: **Data Collection:** Gather $\tilde{n}$ instances of precisely measured data or data with minor mismeasurement, represented as $\mathcal{D} = \left\{ \{\tilde{X}_j, \tilde{Y}_j\} : j = 1, \cdots, \tilde{n} \right\}$, which may come from a previous study.
3: **Dataset Augmentation:** Merge $\mathcal{D}^*$ and $\mathcal{D}$ into an augmented dataset, denoted $\mathcal{D}_A^* \triangleq \mathcal{D}^* \cup \mathcal{D}$.
4: **Output:** Train the optimal classifier using the augmented dataset $\mathcal{D}_A^*$
   4.1. Specify the class $\mathcal{F}$ of classifiers according to a specific application
   4.2. Solve the optimization problem (18) by replacing data $\mathcal{D}$ with $\mathcal{D}_A^*$

---

Table 1: Sensitivity analyses of the CheXpert data using the true (T), naive (N) and the proposed Correction (C) methods.

| Variable | Accuracy (%) | | | Precision (%) | | | Recall (%) | | | F1-score (%) | | |
|---|---|---|---|---|---|---|---|---|---|---|---|---|
| | T | N | C | T | N | C | T | N | C | T | N | C |
| Cardiomegaly | 99.50 | **69.80** | 84.16 | 100 | **58.62** | 100 | 100 | 100 | 78.79 | 99.25 | **58.03** | 69.23 |
| Edema | 99.01 | **76.73** | 94.55 | 100 | **40** | 100 | 95.24 | 100 | 85.71 | 97.56 | **45.71** | 86.75 |
| Consolidation | 100 | **86.63** | 96.04 | 100 | **83.33** | 100 | 100 | 87.50 | 87.50 | 100 | **41.94** | 85.71 |
| Atelectasis | 100 | **73.27** | 88.12 | 100 | 100 | 87.10 | 100 | 100 | 86.67 | 100 | **62.44** | 82.86 |
| Pleural Effusion | 100 | **69.31** | 83.17 | 100 | 100 | 100 | 100 | 82.81 | 82.81 | 100 | **46.29** | 64.58 |

high-quality chest X-ray images from 65,240 patients, annotated for 14 common chest conditions. Like the medical AI competition organized by the Stanford ML group, we aim to train a model to predict the presence or absence for five specific diseases: *Cardiomegaly*, *Edema*, *Consolidation*, *Atelectasis*, and *Pleural Effusion*.

CheXpert provides a validation dataset where labels for the five diseases are considered precise (i.e. noise-free). However, chest X-ray images are inevitably noisy due to various factors like improper patient positioning, suboptimal beam angles or radiologist errors. While the provided images are deemed to be mismeasured, there are no precisely measured images to determine the noise degree.

We conduct sensitivity analyses on the validation data to investigate the impact of noisy inputs and examine the performance of the proposed *correction* method under different measurement error models. For the experiments, we use DenseNet121 (Huang et al. 2017), a convolutional neural network (CNN), as our model architecture with ReLU activation. The *logistic loss*, $\varphi(\epsilon) = \log_2(1 + e^\epsilon)$, is used as the *convex surrogate* function $\varphi(\cdot)$. More implementation details can be found in Appendix D.

The results (Table 1) show that models trained without considering noise (naive method) underperform, while our correction method significantly improves performance, effectively mitigating the impact of noisy inputs.

## 4.2 SYNTHETIC EXPERIMENTS

To investigate the impact of noisy inputs and assess the effectiveness of our *correction* method, we conduct extensive synthetic experiments. We set $n = 1000$ and the input space $\mathcal{X} = \mathbb{R}$. Each configuration, described below, is simulated 100 times. We consider two types of noisy input models: *additive* and *Berkson*, as described in Example C.1 in the appendix.

For the additive noisy input model (C.1), we generate $n$ samples $\left\{\{X_i, X_i^*\} : i = 1, \cdots, n\right\}$ by first drawing the true input $X_i$ for $i = 1, \cdots, n$ independently from the normal distribution with both mean and variance being 1, and then generating the noise term $e_i$ independently for $i = 1, \cdots, n$ from the normal distribution with the mean $\mu$ and variance $\sigma^2$ to determine the noisy input $X_i^*$ of $X_i$.

For the Berkson model (C.2), reverse the process: we first independently generate the noisy input $X_i^*$ from the normal distribution with both mean and variance being 1, and then generate the error term $e_i^*$ for $i = 1, \cdots, n$ independently from the normal distribution with the mean $\mu^*$ and variance $(\sigma^*)^2$ to derive the true input $X_i$ of $X_i^*$.

Next, we generate the label $Y_i$ based on the generated true inputs $X_i$ using a logistic model: $\mathbb{P}(Y_i = 1|X_i) = \sigma(10X_i + 1)$, with the *sigmoid function* $\sigma(u) \triangleq \frac{1}{1+e^{-u}}$, and we independently generate the label $Y_i$ from a *Bernoulli* distribution, with this probability.

To study the impact of varying input noise, we test six different configurations of $(\mu, \sigma)$ for the *additive model*: $(-1, 0.2)$, $(-1.2, 0.2)$, $(-1.4, 0.2)$, $(-1, 0.4)$, $(-1, 0.6)$, $(-1, 0.8)$ (referred to as Cases 1-6). Similarly, for the *Berkson model*, we use six configurations of $(\mu^*, \sigma^*)$: $(-1, 1)$, $(-1.2, 1)$, $(-1.4, 1)$, $(-1, 0.2)$, $(-1, 0.5)$, $(-1, 0.8)$ (called Cases $1^*$-$6^*$, respectively).

For classification, we specify the class $\mathcal{F}$ as the set of all linear functions and take the convex surrogate function $\varphi(\cdot)$ as the *logistic loss*, $\varphi(u) = \log_2(1 + e^u)$. We evaluate three approaches. The true classifier is trained on the data with the true inputs, $\left\{\{X_i, Y_i\} : i = 1, \cdots, n\right\}$; the naive classifier is trained on the noisy inputs $\left\{\{X_i^*, Y_i\} : i = 1, \cdots, n\right\}$; and for the correction method, we train the classifier from the augmented dataset $\mathcal{D}_A^* = \mathcal{D}^* \cup \mathcal{D}$, where $\mathcal{D}^* \triangleq \left\{\{X_i^*, Y_i\} : i = 1, \cdots, n\right\}$, and $\mathcal{D} \triangleq \left\{\{X_j, Y_j\} : j = 1, \cdots, \tilde{n}\right\}$ is additionally independently generated using the same generation process for $\left\{\{X_i, Y_i\} : i = 1, \cdots, n\right\}$. Here, $n = 1,000$ and $K = 10,000$.

For testing, we generate a separate set of 200 precisely measured synthetic samples $\mathcal{T} \triangleq \left\{\{X_k, Y_k\} : k = 1, \cdots, 200\right\}$ by using the preceding data generation process. For each configuration, we report the average values of *accuracy*, *precision*, *recall*, and *F1-score* for predicted labels of the true inputs in the *test* set $\mathcal{T}$ across 100 synthetic datasets to evaluate the performance for the *correction* methods.

Tables 2 and 3 summarize the results for the *additive* and *Berkson models*, respectively. In terms of accuracy and F1-score, the true classifier performs the best, the naive classifier performs the worst, and the proposed *correction* method has similar performance as the *true* method, and these patterns are consistently exhibited under all settings. Regarding precision and recall metrics, the naive method shows extreme variability depending on the noise type. In the additive model, it can achieve 100% recall values but suffers from poor precision values. In contrast, the Berkson model leads to 100% precision values but very low recall values. However, the proposed correction method maintains robust performance, regardless of the input noise form or degree, with the performance close to that of the true method. These findings reveal that the naive method yields unreliable results, and that the proposed correction method effectively mitigates the input noise effects in various settings.

Additional synthetic experiments, exploring the sensitivity to $\tilde{n}$ and misspecification of the input noise model, are deferred to Appendix E.

## 5 DISCUSSION

In this paper, we examine how noisy input affect binary classification and present an informative upper bound on the difference between the *generalization error* and $\varphi$-*risk* of the optimal classifier trained on noisy inputs, compared to the minimum *generalization error* and $\varphi$-*risk* when using precisely measured

Table 2: Synthetic experiment results under the additive model: Performance comparison of the true (T), naive (N), and proposed correction (C) methods.

| Case | Accuracy (%) | | | Precision (%) | | | Recall (%) | | | F1-score (%) | | |
|---|---|---|---|---|---|---|---|---|---|---|---|---|
| | T | N | C | T | N | C | T | N | C | T | N | C |
| 1 | 96.64 | **87.28** | 95.10 | 97.44 | **87.09** | 94.81 | 98.68 | 100 | 99.74 | 98.05 | **93.08** | 97.21 |
| 2 | 96.64 | **86.72** | 94.18 | 97.44 | **86.60** | 93.75 | 98.68 | 100 | 99.88 | 98.05 | **92.80** | 96.71 |
| 3 | 96.64 | **86.35** | 93.08 | 97.44 | **86.27** | 92.57 | 98.68 | 100 | 99.96 | 98.05 | **92.62** | 96.11 |
| 4 | 96.64 | **86.87** | 94.93 | 97.44 | **86.73** | 94.59 | 98.68 | 100 | 99.80 | 98.05 | **92.88** | 97.12 |
| 5 | 96.64 | **86.46** | 94.66 | 97.44 | **86.37** | 94.28 | 98.68 | 100 | 99.83 | 98.05 | **92.67** | 96.97 |
| 6 | 96.64 | **86.09** | 94.22 | 97.44 | **86.05** | 93.80 | 98.68 | 100 | 99.87 | 98.05 | **92.49** | 96.73 |

Table 3: Synthetic experiment results under the Berkson model: Performance comparison of the true (T), naive (N), and proposed correction (C) methods.

| Case | Accuracy (%) | | | Precision (%) | | | Recall (%) | | | F1-score (%) | | |
|---|---|---|---|---|---|---|---|---|---|---|---|---|
| | T | N | C | T | N | C | T | N | C | T | N | C |
| 1* | 96.15 | **73.70** | 95.45 | 96.18 | 100 | 98.37 | 96.44 | **49.47** | 92.79 | 96.29 | **66.02** | 95.48 |
| 2* | 96.22 | **70.83** | 95 | 96.30 | 100 | 98.65 | 95.60 | **37.51** | 90.54 | 95.93 | **54.38** | 94.40 |
| 3* | 96.33 | **70.05** | 94.49 | 95.61 | 100 | 98.97 | 95.42 | **26.95** | 87.49 | 95.48 | **42.19** | 92.83 |
| 4* | 94.69 | **65.44** | 94.32 | 94.76 | 100 | 97.07 | 95.24 | **34.82** | 92.04 | 94.98 | **51.48** | 94.46 |
| 5* | 95.28 | **67.80** | 94.71 | 95.42 | 100 | 97.52 | 95.64 | **39.05** | 92.32 | 95.51 | **56.00** | 94.83 |
| 6* | 95.60 | **71.23** | 95.04 | 95.89 | 100 | 97.95 | 95.74 | **45.12** | 92.50 | 95.79 | **62.00** | 95.12 |

inputs. This upper bound quantifies the effect of input noise, and we show that it diminishes as the noise level decreases. To address the noise issue, we propose a correction method to mitigate the input noise effects by utilizing different model assumptions.

There are interesting directions for future work. One extension is to develop strategies for multiple classification in the presence of input noise. Another important challenge involves cases where both input and output variables are subject to noise, which introduces additional complexities and requires further investigation.

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

**APPENDICES: TECHNICAL DETAILS AND ADDITIONAL EXPERIMENTAL RESULTS**

## A DEFINITIONS OF CONSISTENCY AND $\varphi$-CONSISTENCY

As described in Section 2 of the main text, let $S(n)$ represent a sample of size $n$ consisting of independently and identically distributed (i.i.d.) copies of the input $X$ and output $Y$. The realizations of this sample can be written as $s(n) = \left\{ \{x_1, y_1\}, \cdots, \{x_n, y_n\} \right\}$, which are used as training data to train a classifier, denoted $f_{s(n)}$. This classifier is an element of $\mathcal{F}$, indexed by $s(n)$. Let $f_{S(n)}$ denote a *random* classifier such that, for any realization $s(n)$ of $S(n)$, $f_{S(n)} = f_{s(n)}$.

Essentially, $f_{S(n)}(X)$ depends on both the input $X$ and the sample $S(n)$, introducing two sources of randomness: one from random input variable $X$ and the other from the random sample $S(n)$. In contrast to (5) in the main text, we define $R(f_{S(n)})$ as a random variable that takes the value $R(f_{s(n)})$ when $S(n) = s(n)$, where $R(f_{s(n)})$ is given by (5) in the main text with $f$ replaced by $f_{s(n)}$. That is,

$$R(f_{s(n)}) = \mathbb{E}\{\ell(-Y f_{S(n)}(X)) \big| S(n) = s(n)\}, \tag{A.1}$$

and $R(f_{S(n)})$ remains random due to its dependence on the random sample $S(n)$.

**Definition A.1.** *A sequence of (random) classifiers,* $\{f_{S(n)} : n = 1, 2, \cdots\}$ *is called* consistent *if* $R(f_{S(n)}) \to R_0$ *in probability as the sample size $n$ approaches infinity.*

A similar definition can be found in Biau, Devroye, and Lugosi (2008, p.2017) and Steinwart (2005, p.128). Consistency here refers to the ability of a training method to achieve optimal performance as the sample size $n$ approaches infinity. Similarly, we define the $\varphi$-*consistency* as follows.

**Definition A.2.** *A sequence of classifiers* $\{f_{S(n)} : n = 1, 2, \cdots\}$ *is called $\varphi$-consistent if $R_\varphi(f_{S(n)}) \to R_\varphi(f_0)$ in probability as the sample size $n$ approaches infinity, where $R_\varphi(f_{S(n)})$ is defined similarly to $R(f_{S(n)})$ but with (5) in the main text replaced by (9).*

## B PROOFS OF THEORETICAL RESULTS

### B.1 PROOF OF LEMMA 1

By the fact that any continuous function is bounded over a bounded closed set in $\mathbb{R}$, there exists a constant $E_\varphi > 0$ such that $|\varphi(u)| \le E_\varphi$ for all $u \in [-1, 1]$. Noting that by definition of $f$, $\mathcal{A} \triangleq \{f(x) : f \in \mathcal{F}; x \in \mathcal{X}\}$ is a subset of $[-1, 1]$, we conclude that the image of $\mathcal{A}$ under $\varphi$ is also bounded by $E_\varphi$. That is, we have that $|\varphi(f(x))| \le E_\varphi$ for all $f \in \mathcal{F}$ and $x \in \mathcal{X}$.

### B.2 PROOF OF THEOREM 1

We prove Theorem 1 by the following two steps:

**Step 1**: Proof of (15) in the main text:

First, we find an upper bound of $R_\varphi(\hat{f}_\varphi) - R_\varphi(f_0)$ via $\hat{R}_\varphi(\cdot)$:

$$
\begin{aligned}
R_\varphi(\hat{f}_\varphi) - R_\varphi(f_0) &= R_\varphi(\hat{f}_\varphi) - \hat{R}_\varphi(\hat{f}_\varphi) + \hat{R}_\varphi(\hat{f}_\varphi) - \hat{R}_\varphi(f_0) + \hat{R}_\varphi(f_0) - R_\varphi(f_0) \\
&\le R_\varphi(\hat{f}_\varphi) - \hat{R}_\varphi(\hat{f}_\varphi) + \hat{R}_\varphi(f_0) - R_\varphi(f_0) \\
&\le 2 \sup_{f \in \mathcal{F}} |R_\varphi(f) - \hat{R}_\varphi(f)|, \tag{B.1}
\end{aligned}
$$

where the first inequality is due to $\hat{R}_\varphi(\hat{f}_\varphi) - \hat{R}_\varphi(f_0) \leq 0$ by the definition of $\hat{f}_\varphi$, and the second inequality is due to the property of supremum.

Applying Lemma 1 and repeating the proof of Theorem 4.1 of Boucheron, Bousquet, and Lugosi (2005), we obtain that with probability at least $1 - \delta$,

$$2 \sup_{f \in \mathcal{F}} |R_\varphi(f) - \hat{R}_\varphi(f)| \leq 4L_\varphi \mathcal{R}(\mathcal{F}) + 2E_\varphi \sqrt{\frac{2log(1/\delta)}{n}}. \tag{B.2}$$

Then combining (B.1) and (B.2) proves (15) in the main text.

**Step 2**: Proof of (16) in the main text:

By Theorem 1 and Lemma 2 of Bartlett et al. (2006), for the convex surrogate $\varphi(\cdot)$, there exists a nonnegative continuous convex function $\psi_\varphi(\cdot)$ from $[-1, 1]$ to $\mathbb{R}$ with $\psi_\varphi(0) = 0$ such that

$$\psi_\varphi(R(\hat{f}_\varphi) - R_0) \leq R_\varphi(\hat{f}_\varphi) - \min_{g \in \mathcal{G}} R_\varphi(g). \tag{B.3}$$

If the convex surrogate $\varphi(\cdot)$ is classification-calibration, then by the comment after Theorem 1 of Bartlett et al. (2006), $\psi_\varphi(\cdot)$ is invertible on $[0, 1]$. Thus, we consider the restricted version of $\psi_\varphi(\cdot)$ on $[0, 1]$, and let $\tilde{\psi}_\varphi(\cdot)$ denote it, i.e., $\tilde{\psi}_\varphi(\cdot)$ maps $[0, 1]$ to $\mathbb{R}$ satisfying $\tilde{\psi}_\varphi(x) = \psi_\varphi(x)$ for all $x \in [0, 1]$. Then $\tilde{\psi}_\varphi(\cdot)$ is nonnegative, convex, invertible and continuous over $[0, 1]$, where continuity at the end points 0 and 1 refer to the right-continuous at 0 and left-continuous at 1, respectively. Further, $\tilde{\psi}_\varphi$ is strictly increasing over $[0, 1]$. Indeed, by part 9 of Lemma 2 of Bartlett et al. (2006), for all $x \in (0, 1]$, we have that $\tilde{\psi}_\varphi(x) > 0$, i.e., $\tilde{\psi}_\varphi(x) > \tilde{\psi}_\varphi(0)$ because $\tilde{\psi}_\varphi(0) = \psi_\varphi(0) = 0$; by part 2 of Lemma 1 of Bartlett et al. (2006), we have that for all $0 < y < x \leq 1$, $\tilde{\psi}_\varphi(y) \leq \frac{y}{x}\tilde{\psi}_\varphi(x) < \tilde{\psi}_\varphi(x)$. Therefore, $\tilde{\psi}_\varphi(\cdot)$ is nonnegative, convex, continuous, strictly increasing, and invertible with $\tilde{\psi}_\varphi(0) = 0$.

As the domain $[0, 1]$ of $\tilde{\psi}_\varphi(\cdot)$ is compact and $\mathbb{R}$ is a Hausdorff space (Kelly 2017), by the classical result that the inverse of a continuous bijection from a compact space onto a Hausdorff space is also continuous (Hoffmann 2015), the inverse of $\tilde{\psi}_\varphi(\cdot)$, denoted $\zeta_\varphi(\cdot)$, is continuous. In addition, because $\tilde{\psi}_\varphi(\cdot)$ is strictly increasing with $\tilde{\psi}_\varphi(0) = 0$, its inverse $\zeta_\varphi(\cdot)$ is also strictly increasing with $\tilde{\psi}_\varphi(0) = 0$.

Furthermore, because $R_0$ is the minimum value of $R(h)$ over $\mathcal{H}$ and $\mathcal{F}$ is a subset of $\mathcal{H}$, $0 \leq R(\hat{f}_\varphi) - R_0 \leq 1$. Then by (B.3), we have that

$$\tilde{\psi}_\varphi(R(\hat{f}_\varphi) - R_0) \leq R_\varphi(\hat{f}_\varphi) - \min_{g \in \mathcal{G}} R_\varphi(g).$$

Therefore, by the monotonicity of $\zeta_\varphi(\cdot)$,

$$\begin{aligned}
R(\hat{f}_\varphi) - R_0 &\leq \zeta_\varphi\Big(R_\varphi(\hat{f}_\varphi) - \min_{g \in \mathcal{G}} R_\varphi(g)\Big) \\
&= \zeta_\varphi\Big(\big\{R_\varphi(\hat{f}_\varphi) - R_\varphi(f_0)\big\} + \big\{R_\varphi(f_0) - \min_{g \in \mathcal{G}} R_\varphi(g)\big\}\Big) \\
&\leq \zeta_\varphi\Big(\big\{4L_\varphi \mathcal{R}(\mathcal{F}) + 2E_\varphi \sqrt{\frac{2log(1/\delta)}{n}}\big\} + \big\{R_\varphi(f_0) - \min_{g \in \mathcal{G}} R_\varphi(g)\big\}\Big) \\
&= \zeta_\varphi(D_\varphi), \tag{B.4}
\end{aligned}$$

where the second last step is due to (15) in the main text and monotonicity of $\zeta_\varphi(\cdot)$. That is, (16) in the main text follows.

### B.3 PROOF OF THEOREM 2

The proof of Theorem 2 consists two parts that involves multiple steps.

**Part 1**: Proof of (20) in the main text:

First, we examine $R_\varphi(\hat{f}_\varphi^*) - R_\varphi(f_0)$ via $\hat{R}_\varphi^*(\hat{f}_\varphi^*)$ and $\hat{R}_\varphi^*(f_0)$:

$$
\begin{aligned}
R_\varphi(\hat{f}_\varphi^*) - R_\varphi(f_0) &= \left\{ R_\varphi(\hat{f}_\varphi^*) - \hat{R}_\varphi^*(\hat{f}_\varphi^*) \right\} + \left\{ \hat{R}_\varphi^*(\hat{f}_\varphi^*) - \hat{R}_\varphi^*(f_0) \right\} + \left\{ \hat{R}_\varphi^*(f_0) - R_\varphi(f_0) \right\} \\
&\leq \left\{ R_\varphi(\hat{f}_\varphi^*) - \hat{R}_\varphi^*(\hat{f}_\varphi^*) \right\} + \left\{ \hat{R}_\varphi^*(f_0) - R_\varphi(f_0) \right\} \\
&\leq 2 \sup_{f \in \mathcal{F}} \left| R_\varphi(f) - \hat{R}_\varphi^*(f) \right| \\
&= 2 \sup_{f \in \mathcal{F}} \left| \left\{ R_\varphi(f) - \hat{R}_\varphi(f) \right\} + \left\{ \hat{R}_\varphi(f) - \hat{R}_\varphi^*(f) \right\} \right| \\
&\leq 2 \sup_{f \in \mathcal{F}} \left| R_\varphi(f) - \hat{R}_\varphi(f) \right| + 2 \sup_{f \in \mathcal{F}} \left| \hat{R}_\varphi(f) - \hat{R}_\varphi^*(f) \right|,
\end{aligned}
$$

where the first inequality is due to $\hat{R}_\varphi^*(\hat{f}_\varphi^*) - \hat{R}_\varphi^*(f_0) \leq 0$ by the definition of $\hat{f}_\varphi^*$ in (18) in the main text, the second inequality comes from the property of supremum and the fact that $\hat{f}_\varphi^* \in \mathcal{F}$, and the last inequality is due to the triangle inequality of absolute value.

Therefore,

$$
\mathbb{E}\left\{ R_\varphi(\hat{f}_\varphi^*) - R_\varphi(f_0) \right\} \leq \mathbb{E}\left\{ 2 \sup_{f \in \mathcal{F}} \left| R_\varphi(f) - \hat{R}_\varphi(f) \right| \right\} + \mathbb{E}\left\{ 2 \sup_{f \in \mathcal{F}} \left| \hat{R}_\varphi(f) - \hat{R}_\varphi^*(f) \right| \right\}. \tag{B.5}
$$

Now we examine the two terms in (B.5) separately in the following two steps.

**Step 1**: Examine $\mathbb{E}\left\{ 2 \sup_{f \in \mathcal{F}} \left| R_\varphi(f) - \hat{R}_\varphi(f) \right| \right\}$ in (B.5):

For any $f \in \mathcal{F}$, we have that

$$
\begin{aligned}
\left| R_\varphi(f) - \hat{R}_\varphi(f) \right| &\leq \left| R_\varphi(f) \right| + \left| \hat{R}_\varphi(f) \right| \\
&= \left| \mathbb{E}\{\varphi(-Yf(X))\} \right| + \left| \frac{1}{n} \sum_{i=1}^n \varphi(-Y_i f(X_i)) \right| \\
&\leq \mathbb{E}\left| \varphi(-Yf(X)) \right| + \frac{1}{n} \sum_{i=1}^n \left| \varphi(-Y_i f(X_i)) \right| \\
&\leq E_\varphi + \frac{1}{n} \sum_{i=1}^n E_\varphi \\
&= 2E_\varphi, \tag{B.6}
\end{aligned}
$$

where the first step is due to the triangle inequality of absolute value, the second step is due to (9) and (11) in the main text, the third step is due to Jensen's inequality, and the fourth step is due to Lemma 1 in the main text.

Next, applying (B.2) to the case with $\delta = \frac{1}{n}$ and using (13) in the main text, we obtain that

$$
\mathbb{P}\left( 2 \sup_{f \in \mathcal{F}} |R_\varphi(f) - \hat{R}_\varphi(f)| > C(n, \frac{1}{n}) \right) \leq \frac{1}{n}. \tag{B.7}
$$

Consequently,

$$\mathbb{E}\Big\{2\sup_{f\in\mathcal{F}}|R_\varphi(f)-\hat{R}_\varphi(f)|\Big\}$$

$$=\mathbb{E}\Big[\big\{2\sup_{f\in\mathcal{F}}|R_\varphi(f)-\hat{R}_\varphi(f)|\big\}\mathbb{1}_{\big\{2\sup_{f\in\mathcal{F}}|R_\varphi(f)-\hat{R}_\varphi(f)|\le C(n,\frac{1}{n})\big\}}$$

$$+\big\{2\sup_{f\in\mathcal{F}}|R_\varphi(f)-\hat{R}_\varphi(f)|\big\}\mathbb{1}_{\big\{2\sup_{f\in\mathcal{F}}|R_\varphi(f)-\hat{R}_\varphi(f)|>C(n,\frac{1}{n})\big\}}\Big]$$

$$\le\mathbb{E}\Big[C(n,\frac{1}{n})\mathbb{1}_{\big\{2\sup_{f\in\mathcal{F}}|R_\varphi(f)-\hat{R}_\varphi(f)|\le C(n,\frac{1}{n})\big\}}+4E_\varphi\mathbb{1}_{\big\{2\sup_{f\in\mathcal{F}}|R_\varphi(f)-\hat{R}_\varphi(f)|>C(n,\frac{1}{n})\big\}}\Big]$$

$$=C(n,\frac{1}{n})\mathbb{P}\Big(2\sup_{f\in\mathcal{F}}|R_\varphi(f)-\hat{R}_\varphi(f)|\le C(n,\frac{1}{n})\Big)+4E_\varphi\mathbb{P}\Big(2\sup_{f\in\mathcal{F}}|R_\varphi(f)-\hat{R}_\varphi(f)|>C(n,\frac{1}{n})\Big)$$

$$\le C(n,\frac{1}{n})+\frac{4E_\varphi}{n}, \tag{B.8}$$

where the first inequality is due to the property of indicator function and (B.6), the second equality is due to the property of indicator function, and the last inequality is due to (B.7) and the fact that the probability is always less than or equal to 1.

**Step 2**: Examine $\mathbb{E}\Big\{2\sup_{f\in\mathcal{F}}\big|\hat{R}_\varphi(f)-\hat{R}_\varphi^*(f)\big|\Big\}$ in (B.5):

By (11) and (18) in the main text, we obtain that

$$2\sup_{f\in\mathcal{F}}|\hat{R}_\varphi(f)-\hat{R}_\varphi^*(f)|$$

$$=2\sup_{f\in\mathcal{F}}\Big|\frac{1}{n}\sum_{i=1}^n\varphi(-Y_if(X_i))-\frac{1}{n}\sum_{i=1}^n\varphi(-Y_if(X_i^*))\Big|$$

$$\le 2\sup_{f\in\mathcal{F}}\frac{1}{n}\sum_{i=1}^n\Big|\varphi(-Y_if(X_i))-\varphi(-Y_if(X_i^*))\Big|$$

$$\le\frac{2}{n}\sum_{i=1}^n\sup_{f\in\mathcal{F}}\Big|\varphi(-Y_if(X_i))-\varphi(-Y_if(X_i^*))\Big|$$

$$\le\frac{2}{n}\sum_{i=1}^n\sup_{f\in\mathcal{F}}\Big[L_\varphi\big|Y_i\cdot\{f(X_i)-f(X_i^*)\}\big|\Big]$$

$$=\frac{2L_\varphi}{n}\sum_{i=1}^n\sup_{f\in\mathcal{F}}\big|f(X_i)-f(X_i^*)\big|$$

$$\le\frac{2L_\varphi L_\mathcal{F}}{n}\sum_{i=1}^n\sup_{f\in\mathcal{F}}||X_i-X_i^*||_2, \tag{B.9}$$

where the first inequality is due to the property of absolute value, the second inequality is due to the property of supremum, the third inequality is due to (7) in the main text, the second last step is due to that $Y_i\in\{-1,1\}$ for any $i$, and the last step is due to (19) in the main text.

By taking expectation on both sides of (B.9) with utilizing (17) in the main text and Jensen's inequality, we obtain that

$$\mathbb{E}\Big\{2\sup_{f\in\mathcal{F}}|\hat{R}_\varphi(f)-\hat{R}_\varphi^*(f)|\Big\} \leq \frac{2L_\varphi L_\mathcal{F}}{n}\sum_{i=1}^{n}\sqrt{D_i}. \tag{B.10}$$

Consequently, applying (B.10) and (B.8) to (B.5) proves (20) in the main text.

**Part 2**: Proof of (21) in the main text.

Repeat the proof for (16) in the main text presented in Step 2 of B.2, with $\hat{f}_\varphi$ replaced by $\hat{f}_\varphi^*$, where $\varphi$ is assumed to be classification-calibrated. Then we can show that there exists a nonnegative, convex, continuous, strictly increasing, and invertible function, denoted $\tilde{\psi}_\varphi(\cdot)$, such that

$$\tilde{\psi}_\varphi(R(\hat{f}_\varphi^*)-R_0) \leq R_\varphi(\hat{f}_\varphi^*) - \min_{g\in\mathcal{G}}R_\varphi(g),$$

and the function $\tilde{\psi}_\varphi(\cdot)$ has the following properties:

   (a). $\tilde{\psi}_\varphi(0) = 0$;

   (b). its inverse function, denoted $\zeta_\varphi(\cdot)$, is continuous and satisfies $\zeta_\varphi(0) = 0$.

Then by Jensen's inequality, we obtain that

$$\tilde{\psi}_\varphi\Big(\mathbb{E}\big\{R(\hat{f}_\varphi^*)-R_0\big\}\Big) \leq \mathbb{E}\big\{\tilde{\psi}_\varphi(R(\hat{f}_\varphi^*)-R_0)\big\} \leq \mathbb{E}\Big\{R_\varphi(\hat{f}_\varphi^*)-\min_{g\in\mathcal{G}}R_\varphi(g)\Big\}. \tag{B.11}$$

Therefore, by the monotonicity of $\zeta_\varphi(\cdot)$,

$$\mathbb{E}\big\{R(\hat{f}_\varphi^*)-R_0\big\} \leq \zeta_\varphi\Big(\mathbb{E}\Big\{R_\varphi(\hat{f}_\varphi^*)-\min_{g\in\mathcal{G}}R_\varphi(g)\Big\}\Big)$$

$$= \zeta_\varphi\Big(\mathbb{E}\Big[\big\{R_\varphi(\hat{f}_\varphi^*)-R_\varphi(f_0)\big\} + \big\{R_\varphi(f_0)-\min_{g\in\mathcal{G}}R_\varphi(g)\big\}\Big]\Big)$$

$$= \zeta_\varphi\Big[\mathbb{E}\big\{R_\varphi(\hat{f}_\varphi^*)-R_\varphi(f_0)\big\} + \big\{R_\varphi(f_0)-\min_{g\in\mathcal{G}}R_\varphi(g)\big\}\Big]$$

$$\leq \zeta_\varphi\Big(\frac{4E_\varphi}{n} + \frac{2L_\varphi L_\mathcal{F}}{n}\sum_{i=1}^{n}\sqrt{D_i} + D_\varphi(n,\frac{1}{n})\Big), \tag{B.12}$$

where the first inequality is due to (B.11), and the last inequality is due to (14) in the main text, (20) in the main text, and monotonicity of $\zeta_\varphi(\cdot)$. That is, (21) in the main text follows.

## C   EXAMPLES OF INPUT NOISE MODELS

**Example C.1.** *Suppose we have $X_i$ and $X_i^*$ defined by one of the following four common input noise models.*

   *(1) Additive model:*

$$X_i^* = X_i + e_i, \tag{C.1}$$

   *where $e_i$ is a noise term with zero mean and covariance matrix $\Sigma_i$, and is independent of $X_i$. In this case, we can compute the expected squared difference between $X_i$ and $X_i^*$ as:*

   $$D_i = \mathbb{E}\{||X_i^*-X_i||_2^2\} = \mathbb{E}\{e_i^T e_i\} = \mathbb{E}\{tr(e_i^T e_i)\} = \mathbb{E}\{tr(e_i e_i^T)\} = tr(\mathbb{E}\{e_i e_i^T\}) = tr(\Sigma_i),$$

   *where $tr(\cdot)$ denotes the trace of a matrix, the second equality is due to the fact that $e_i^T e_i$ is a scalar, the third equality is due to the property that $tr(AB) = tr(BA)$ for any matrices $A$ and $B$, and the fourth equality is due to the fact that $tr(\cdot)$ is a linear operator.*

*(2) Berkson model:*

$$X_i = X_i^* + e_i^*, \tag{C.2}$$

*where $e_i^*$ is noise with zero mean and covariance matrix $\Sigma_i^*$, and is independent of $X_i^*$. Similarly, the expected squared difference is given by:*

$$D_i = \mathbb{E}\{||X_i^* - X_i||_2^2\} = \mathbb{E}\{e_i^{*T}e_i^*\} = \mathbb{E}\{tr(e_i^{*T}e_i^*)\} = \mathbb{E}\{tr(e_i^*e_i^{*T})\} = tr(\mathbb{E}\{e_i^*e_i^{*T}\}) = tr(\Sigma_i^*).$$

*(3) Multiplicative model: $X_i^* = X_i e_i$, where $e_i$ is a scalar noise term with mean 1 and variance $\sigma_i^2$ and is independent of $X_i$.*

*For $i = 1, \cdots, n$, let $\mu_{2i} = \mathbb{E}(X_i^T X_i)$. Then the expected squared difference is*

$$\begin{aligned} D_i = \mathbb{E}\{||X_i^* - X_i||_2^2\} &= \mathbb{E}\{(e_i - 1)^2 X_i^T X_i\} \\ &= \mathbb{E}\{(e_i - 1)^2\}\mathbb{E}\{X_i^T X_i\} \\ &= \sigma_i^2 \mu_{2i}, \end{aligned}$$

*where the second equality is due to the assumption that $e_i$ is independent of $X_i$.*

*(4) Berkson-type multiplicative model: $X_i = X_i^* e_i^*$, where $e_i^*$ is a scalar noise term with mean 1 and variance $\sigma_i^{*2}$ and $e_i^*$ is independent of $X_i^*$.*

*For $i = 1, \cdots, n$, let $\mu_{2i}^* = \mathbb{E}(X_i^{*T} X_i^*)$. Then the expected squared difference becomes*

$$\begin{aligned} D_i = \mathbb{E}\{||X_i^* - X_i||_2^2\} &= \mathbb{E}\{(e_i^* - 1)^2 X_i^{*T} X_i^*\} \\ &= \mathbb{E}\{(e_i^* - 1)^2\}\mathbb{E}\{X_i^{*T} X_i^*\} \\ &= \sigma_i^{*2} \mu_{2i}^*, \end{aligned}$$

*where the second equality is due to the assumption that $e_i^*$ is independent of $X_i^*$.*

In each case, the expected error $D_i$ quantifies how much the input noise distorts the data, depending on the model assumption. This formulation covers a range of scenarios, from simple additive noise to multiplicative distortions, commonly used in the literature (Yi 2017; Yi, Delaigle, and Gustafson 2021).

## D  IMPLEMENTATION IN MEDICAL IMAGE DATA ANALYSIS

The sensitivity analyses proceed in the following four steps:

1. Image Preprocessing: We apply the codes from Irvin et al. (2019) to process each image into a $3 \times 224 \times 224$ array. This preprocessing includes random rotation, translation, and scaling. Let $X_i^* \triangleq \{X_{ijk}^* \in \mathbb{R}^3 : j, k = 1, \cdots, 224\}$ denote the $3 \times 224 \times 224$ array corresponding to the $i$th noisy image.

2. Generate Precise Measurements: We create a precisely measured version of the image, $X_{ijk}$, independently using the *Berkson* model described in Example C.1:

$$X_{ijk} = X_{ijk}^* + e_{ijk}^*, \tag{D.1}$$

where $e_{ijk}^*$ is drawn from a normal distribution with mean $a_{ijk}$ and an identity matrix as the covariance matrix. Here, we specify $a_{ijk}$ as $(0.8, 0.7, 0.6)^{\mathrm{T}}$.

3. Training Classifiers: We train the *true* classifier from the generated precisely measured inputs and their corresponding outputs, as described in Section 2. The *true* classifier, denoted as $\hat{f}_\varphi$, is obtained by solving the optimization problem (12) in the main text. In contrast, the *naive* and the proposed *corrected* classifiers are trained using the noisy inputs and the outputs, as described in Section 3. The *naive* classifier, $\hat{f}_\varphi^*$, is obtained by solving the optimization problems (18) in the main text. For the proposed correction method, we randomly select $200 - \tilde{n}$ noisy images from a total of 200 noisy images along with their corresponding outputs to create the observed data $\mathcal{D}^*$. The precisely measured versions of the remaining noisy images and their outputs serve as the historical dataset $\mathcal{D}$. We set $\tilde{n} = 160$. We use Adam (Kingma and Ba 2015), a widely used stochastic optimization method for training neural networks, with a batch size of 10.

4. Evaluation Metrics: Finally, for each of the five selected diseases, we calculate the *accuracy*, *precision*, *recall*, and *F1-score* for the *true*, *naive*, and the proposed *corrected* classifiers across the 200 generated precisely measured inputs, respectively.

# E  ADDITIONAL SYNTHETIC EXPERIMENTS

In Section 4.2 of the main text, we conduct synthetic experiments. In addition to those experiments, we want to evaluate how the performance of the proposed correction method may change with the size $\tilde{n}$ of the dataset $\mathcal{D}$. In Tables E.1 and E.2, we report the results for the *additive* and *Berkson models* (C.1) and (C.2), respectively, where we examine $\tilde{n} = 5000, 10000, 20000, 30000,$ and $40000$. The results show that the proposed correction method maintain stable performance with varying values of $\tilde{n}$, although we observe some variations in the performance.

Table E.1: Results of synthetic experiment assessing the impact of $\tilde{n}$ on the proposed correction method: Additive input noise model (C.1).

| Case | Accuracy (%) | | | | | Precision (%) | | | | |
|---|---|---|---|---|---|---|---|---|---|---|
| | 5000 | 10000 | 20000 | 30000 | 40000 | 5000 | 10000 | 20000 | 30000 | 40000 |
| 1 | 92.98 | 95.10 | 96.16 | 96.49 | 96.63 | 92.47 | 94.82 | 96.32 | 96.94 | 97.22 |
| 2 | 91.62 | 94.18 | 95.74 | 96.26 | 96.46 | 91.11 | 93.75 | 95.70 | 96.48 | 96.92 |
| 3 | 90.05 | 93.08 | 95.29 | 95.96 | 96.31 | 89.61 | 92.57 | 95.07 | 96.05 | 96.57 |
| 4 | 92.67 | 94.93 | 96.10 | 96.41 | 96.6 | 92.14 | 94.59 | 96.22 | 96.84 | 97.16 |
| 5 | 92.01 | 94.66 | 95.99 | 96.37 | 96.56 | 91.49 | 94.28 | 96.07 | 96.72 | 97.08 |
| 6 | 91.30 | 94.22 | 95.82 | 96.27 | 96.51 | 90.80 | 93.80 | 95.81 | 96.54 | 96.98 |

| Case | Recall (%) | | | | | F1-score (%) | | | | |
|---|---|---|---|---|---|---|---|---|---|---|
| | 5000 | 10000 | 20000 | 30000 | 40000 | 5000 | 10000 | 20000 | 30000 | 40000 |
| 1 | 99.96 | 99.74 | 99.32 | 99.04 | 98.90 | 96.06 | 97.21 | 97.79 | 97.97 | 98.05 |
| 2 | 99.99 | 99.88 | 99.50 | 99.25 | 99.02 | 95.33 | 96.71 | 97.56 | 97.84 | 97.95 |
| 3 | 100 | 99.96 | 99.67 | 99.38 | 99.21 | 94.51 | 96.11 | 97.31 | 97.68 | 97.87 |
| 4 | 99.98 | 99.80 | 99.35 | 99.05 | 98.93 | 95.89 | 97.12 | 97.76 | 97.93 | 98.03 |
| 5 | 99.99 | 99.83 | 99.39 | 99.13 | 98.97 | 95.54 | 96.97 | 97.70 | 97.91 | 98.01 |
| 6 | 99.99 | 99.87 | 99.48 | 99.21 | 99.01 | 95.16 | 96.73 | 97.60 | 97.85 | 97.98 |

Table E.2: Results of synthetic experiment assessing the impact of $\tilde{n}$ on the proposed correction method: Berkson input noise model (C.2).

| Case | Accuracy (%) | | | | | Precision (%) | | | | |
|---|---|---|---|---|---|---|---|---|---|---|
| | 5000 | 10000 | 20000 | 30000 | 40000 | 5000 | 10000 | 20000 | 30000 | 40000 |
| 1* | 94.03 | 95.45 | 96.02 | 96.04 | 96.10 | 99.29 | 98.37 | 97.55 | 97.02 | 96.85 |
| 2* | 92.56 | 95 | 95.82 | 95.96 | 96.03 | 99.43 | 98.65 | 97.66 | 97.23 | 96.98 |
| 3* | 91.66 | 94.49 | 95.77 | 96.02 | 96.11 | 99.73 | 98.97 | 97.82 | 97.23 | 96.82 |
| 4* | 93.15 | 94.32 | 94.59 | 94.65 | 94.65 | 98.33 | 97.07 | 96.10 | 95.69 | 95.42 |
| 5* | 93.23 | 94.7 | 95.2 | 95.31 | 95.35 | 98.56 | 97.52 | 96.70 | 96.38 | 96.19 |
| 6* | 93.65 | 95.04 | 95.48 | 95.51 | 95.52 | 98.97 | 97.95 | 97.13 | 96.73 | 96.51 |

| Case | Recall (%) | | | | | F1-score (%) | | | | |
|---|---|---|---|---|---|---|---|---|---|---|
| | 5000 | 10000 | 20000 | 30000 | 40000 | 5000 | 10000 | 20000 | 30000 | 40000 |
| 1* | 89.18 | 92.80 | 94.72 | 95.32 | 95.62 | 93.93 | 95.48 | 96.10 | 96.14 | 96.21 |
| 2* | 84.57 | 90.54 | 93.28 | 94.05 | 94.43 | 91.37 | 94.40 | 95.40 | 95.59 | 95.67 |
| 3* | 79.90 | 87.48 | 91.74 | 92.93 | 93.58 | 88.66 | 92.82 | 94.65 | 95.01 | 95.14 |
| 4* | 88.59 | 92.05 | 93.58 | 94.14 | 94.44 | 93.18 | 94.47 | 94.80 | 94.89 | 94.91 |
| 5* | 88.47 | 92.30 | 94.12 | 94.66 | 94.94 | 93.22 | 94.81 | 95.37 | 95.49 | 95.54 |
| 6* | 88.82 | 92.50 | 94.17 | 94.65 | 94.91 | 93.59 | 95.12 | 95.60 | 95.65 | 95.68 |

Table E.3: Results of synthetic experiment results assessing the sensitivity of the proposed correction method to misspecification of the input noise model: Average values of accuracy (%), precision (%), recall (%), and F1-score (%), with the additive input noise model (C.1) being the true model.

| Situation | Accuracy | | | | Precision | | | |
|---|---|---|---|---|---|---|---|---|
| Case | 1 | 2 | 3 | 4 | 1 | 2 | 3 | 4 |
| 1 | 95.10 | 95.10 | 95.10 | 95.10 | 94.82 | 94.82 | 94.82 | 94.82 |
| 2 | 94.18 | 94.18 | 94.18 | 94.18 | 93.75 | 93.75 | 93.75 | 93.75 |
| 3 | 93.08 | 93.08 | 93.08 | 93.08 | 92.57 | 92.57 | 92.57 | 92.57 |
| 4 | 94.93 | 94.93 | 94.66 | 94.93 | 94.59 | 94.59 | 94.28 | 94.59 |
| 5 | 94.66 | 94.66 | 94.66 | 94.66 | 94.28 | 94.28 | 94.28 | 94.28 |
| 6 | 94.22 | 94.22 | 94.22 | 94.22 | 93.80 | 93.80 | 93.80 | 93.80 |

| Situation | Recall | | | | F1-score | | | |
|---|---|---|---|---|---|---|---|---|
| Case | 1 | 2 | 3 | 4 | 1 | 2 | 3 | 4 |
| 1 | 99.74 | 99.74 | 99.74 | 99.74 | 97.21 | 97.21 | 97.21 | 97.21 |
| 2 | 99.88 | 99.88 | 99.88 | 99.88 | 96.71 | 96.71 | 96.71 | 96.71 |
| 3 | 99.96 | 99.96 | 99.96 | 99.96 | 96.11 | 96.11 | 96.11 | 96.11 |
| 4 | 99.80 | 99.80 | 99.80 | 99.80 | 97.12 | 97.12 | 97.12 | 97.12 |
| 5 | 99.83 | 99.83 | 99.83 | 99.83 | 96.97 | 96.97 | 96.97 | 96.97 |
| 6 | 99.87 | 99.87 | 99.87 | 99.87 | 96.73 | 96.73 | 96.73 | 96.73 |

The effectiveness of the proposed correction methods hinges on the knowledge of the input noise model. To assess how the proposed methods perform when the model is misspecified, we conduct sensitivity analyses.

We generate two datasets: precise dataset $\{(X_i, Y_i) : i = 1, \cdots, n\}$ and the noisy dataset $\{(X_i^*, Y_i) : i = 1, \cdots, n\}$ by repeating the data generation procedure described in Section 4.2 of the main text. To implement the proposed correction method, we intentionally misspecify the mean and variance of $e_i$ in the *additive model* (C.1) as $\mu + a_1$ and $(\sigma + a_2)^2$, respectively. We consider four scenarios for $(a_1, a_2)$: $(0, 0)$, $(-0.2, 0.1)$, $(0.2, -0.1)$, and $(-0.2, -0.1)$, called Situations 1-4, respectively. Similarly, in the *Berkson model* (C.2), we misspecify the mean and variance of $e_i^*$ as $\mu^* + a_1^*$ and $(\sigma^* + a_2^*)^2$, respectively. The same pairs for $(a_1, a_2)$ are used for $(a_1^*, a_2^*)$, leading to Situations $1^* - 4^*$.

Situation 1 (or $1^*$) represents the scenario with no input noise, while the other situations illustrate different model misspecification scenarios. For the proposed correction method, we set $\tilde{n} = 10,000$.

Table E.3 presents the average values of accuracy, precision, recall, and F1-score of the proposed *correction* method across Cases 1-6 for Situations 1-4 in the *additive model*. Similarly, Table E.4 displays the results for the *Berkson model*. The performance of the proposed *correction* method is similar across the four selected different values of $(a_1, a_2)$ or $(a_1^*, a_2^*)$ in each case, demonstrating the robustness of the proposed *correction* method against misspecification of the input noise model.

Table E.4: Results of synthetic experiment results assessing the sensitivity of the proposed correction method to misspecification of the input noise model: Average values of accuracy (%), precision (%), recall (%), and F1-score (%), with the Berkson input noise model (C.2) being the true model.

| Situation | Accuracy | | | | Precision | | | |
|---|---|---|---|---|---|---|---|---|
| Case | 1 | 2 | 3 | 4 | 1 | 2 | 3 | 4 |
| 1* | 95.45 | 95.21 | 95.62 | 95.36 | 98.37 | 98.55 | 98.21 | 98.44 |
| 2* | 95 | 94.60 | 95.21 | 94.71 | 98.65 | 98.82 | 98.42 | 98.74 |
| 3* | 94.49 | 94.07 | 94.86 | 94.14 | 98.97 | 99.17 | 98.63 | 99.17 |
| 4* | 94.32 | 94.19 | 94.44 | 94.25 | 97.07 | 97.34 | 96.90 | 97.27 |
| 5* | 94.7 | 94.52 | 94.82 | 94.61 | 97.52 | 97.70 | 97.35 | 97.62 |
| 6* | 95.04 | 94.86 | 95.19 | 94.97 | 97.95 | 98.08 | 97.78 | 98.01 |
| Situation | Recall | | | | F1-score | | | |
| Case | 1 | 2 | 3 | 4 | 1 | 2 | 3 | 4 |
| 1* | 92.80 | 92.15 | 93.29 | 92.55 | 95.48 | 95.23 | 95.67 | 95.39 |
| 2* | 90.54 | 89.52 | 91.22 | 89.82 | 94.40 | 93.92 | 94.66 | 94.05 |
| 3* | 87.48 | 86.27 | 88.71 | 86.45 | 92.82 | 92.22 | 93.36 | 92.32 |
| 4* | 92.05 | 91.53 | 92.45 | 91.70 | 94.47 | 94.32 | 94.60 | 94.38 |
| 5* | 92.30 | 91.76 | 92.70 | 92.29 | 94.81 | 94.61 | 94.94 | 94.72 |
| 6* | 92.50 | 92.02 | 92.96 | 92.29 | 95.12 | 94.92 | 95.04 | 95.04 |

