# OpenReview forum: "Mitigating Input Noise in Binary Classification: A Unified Framework with Data Augmentation"
_ICLR.cc/2025/Conference — ICLR 2025 Conference Withdrawn Submission_

### Official Review · Reviewer_CPGL · 2024-10-29

**Soundness:** 3
**Presentation:** 3
**Contribution:** 2
**Rating:** 6
**Confidence:** 2

**Summary:**

The paper proposes a novel framework for binary classification that involves data with feature noise. The framework uses a convex surrogate loss function for optimisation and a dataset augmentation strategy that mixes the precisely measured data and noisy data. The authors provide a comprehensive theoretical analysis of input noise, demonstrating the bias introduced when noise is ignored, and establishing an upper bound on the disparity between the generalization error and the $\phi$ risk of the optimal classifier, which diminishes as the noise level decreases. Additionally, they introduce a correction method based on data augmentation, generating data with minimal error to create more robust classifiers. The proposed approach is model-agnostic, extending its applicability to various classification algorithms and offering flexibility across different usages.

**Strengths:**

1. The paper provides solid theoretical validation for the generalization error bound when using surrogate convex losses with noisy data.
2. The presentation of the paper is clear, with the proofs and explanations easy to follow.

**Weaknesses:**

1. Although the authors provide solid theoretical analysis of the problem, the proved theorems and lemmas, except for Theorem 2, are considered incremental to existing generalization bound proofs.
2. The authors have not clearly explained the necessity of injecting clean data into the dataset, as this is not evident from the proofs. It appears from Eq 20 and Eq 21 that injecting noisy data might also suffice.
3. While the paper offers theoretical analysis, the practical impact of the proposed method is limited, as collecting a large amount of precisely measured data can be costly in real-world applications.

**Questions:**

See Weaknesses Above.

---

### Official Review · Reviewer_k8tH · 2024-11-03

**Soundness:** 3
**Presentation:** 3
**Contribution:** 2
**Rating:** 3
**Confidence:** 3

**Summary:**

This paper addresses binary classification with noisy input features by introducing a theoretical framework to analyze noise impact on classifier performance and proposing a correction method based on data augmentation. The authors provide theoretical bounds for noise effects and validate their approach through medical image classification and synthetic experiments, supporting their theoretical findings with mathematical proofs and empirical results.

**Strengths:**

1. The paper presents a rigorous theoretical analysis with comprehensive mathematical foundations that clearly establish bounds for input noise impact on classification performance.

2. The proposed framework's model-agnostic nature makes it applicable to various classification algorithms, demonstrating its versatility.

3. The authors provide detailed theoretical proofs in the appendix, and their systematic analysis of noise effects through sensitivity studies adds credibility to their findings.

**Weaknesses:**

1. The proposed correction method is notably simplistic and lacks novelty. While simplicity can be valuable, the authors could have explored more sophisticated approaches to enhance the practical utility. For example:
- Investigating adaptive weighting schemes between noisy and clean data
- Exploring progressive data augmentation strategies
- Considering domain adaptation techniques when clean and noisy data distributions differ

2. The experimental validation is severely limited in scope and diversity. The authors only utilize one real-world dataset (CheXpert) with a single type of noise model, and their synthetic experiments employ basic noise models (additive and Berkson). The lack of diverse datasets and noise scenarios, absence of comparisons with existing methods, and limited exploration of different clean data quantities significantly weaken the paper's empirical contributions. To strengthen the empirical evaluation, the authors should:
- Include additional real-world datasets (e.g., CIFAR-10, ImageNet)
- Experiment with more realistic noise models (e.g., corruption from image compression, or sensor noise)
- Compare with existing noise-handling methods such as robust training algorithms, noise-aware loss functions, or data cleaning techniques
- Conduct ablation studies with varying ratios of clean to noisy data

3. The paper suffers from inconsistent notation and insufficient experimental details. Critical information is missing, such as:
- Training hyperparameters (learning rate, optimizer settings, batch size)
- Validation protocol and split ratios
- Computational resources and training time
- Statistical analysis (e.g., standard deviations, confidence intervals) of the experimental results
- Criteria for determining the optimal amount of clean data

**Questions:**

1. In Table 1, what do the bold numbers represent? The criteria for highlighting certain results are not explained.

2. In Section 4.2, there is confusion about the notation K=10000. Is this the same as ñ from Algorithm 1? Why use different notation?

3. While the theoretical analysis focuses on binary classification, wouldn't the proposed method naturally extend to multi-class problems?  The performance improvement from adding clean data seems independent of the number of classes.

4. How practical is the assumption of having access to clean data? The paper doesn't discuss the challenges or costs of obtaining such data in real-world scenarios.

5. Have the authors considered comparing their method with other noise-handling approaches from the literature?

---

### Official Review · Reviewer_RUAY · 2024-11-04

**Soundness:** 3
**Presentation:** 2
**Contribution:** 3
**Rating:** 6
**Confidence:** 3

**Summary:**

This paper presents a comprehensive framework for binary classification that effectively addresses noisy inputs and is applicable to various supervised learning algorithms and noise models. It explores the biases that can arise when input noise is ignored, highlighting cases where this oversight may be acceptable. For cases that require noise correction, the paper introduces a data augmentation technique designed to minimize the impact of noise, supported by theoretical insights and practical results.

**Strengths:**

1. Addressing binary classification with noisy input is an impactful problem.
2. The experiments conducted with the proposed methods are adequate.

**Weaknesses:**

1. While the paper presents robust synthetic evaluations, real-world performance may differ. Testing the proposed method on additional domains or a broader range of real-world datasets could be beneficial.

   [1] The FIX Benchmark: Extracting Features Interpretable to eXperts

   [2] ABO: Dataset and Benchmarks for Real-World 3D Object Understanding

2. Providing a detailed comparison or citing specific studies that utilize similar augmentation or model-agnostic approaches as baselines could help highlight the novelty of the work.
3. It would be advantageous to conduct more experiments that address the research question. To demonstrate the method's generalizability, it would be useful to test with various models, such as Vision Transformers (ViT).

   [3] An Image is Worth 16x16 Words: Transformers for Image Recognition at Scale

   [4] Learning Transferable Visual Models From Natural Language Supervision
4. Including results under more complex input noise models, such as those involving multiplicative distortions, would provide a more comprehensive evaluation.

**Questions:**

Please see Weaknesses.

---

### Official Review · Reviewer_gssV · 2024-11-04

**Soundness:** 2
**Presentation:** 1
**Contribution:** 1
**Rating:** 3
**Confidence:** 3

**Summary:**

This paper presents a unified framework for handling noisy inputs in binary classification problems, addressing a significant gap in machine learning literature where most existing work focuses on noisy labels rather than noisy features. The authors develop theoretical foundations for understanding when input noise can be safely ignored, propose a data augmentation-based correction method, and validate their approach through experiments on medical images and synthetic data.

**Strengths:**

1. Theoretical Rigor:
The paper provides a comprehensive theoretical analysis with proofs.

2. Practical Relevance:
  The paper addresses a real-world problem (noisy inputs) that affects many applications and proposes data augmentation techniques.

**Weaknesses:**

1. Paper presentation: the paper presentation needs to be improved. For instance, 3 pages out of 10 for the background section (section 2) is too long.

2. Limited Empirical Validation:
My main concern with this paper is the empirical validation. Experiments are severely limited with no baselines. Authors need to compare their methods with adversarial robustness methods. Furthermore, the authors should provide an ablation study.

3. Theoretical Assumptions
The method assumes the availability of less noisy or clean data for augmentation. The method may not fully account for complex, non-uniform noise patterns. Since it relies on convex surrogate functions, it might not always approximate well and may be restrictive in practice.

**Questions:**

1. What happens when noise patterns are heteroscedastic or feature-dependent?

2. How does the method perform with different types of data augmentation?

3. How would the approach handle structured noise in sequential data?

4. How does the method compare to existing denoising techniques?

---

### Official Review · Reviewer_19SA · 2024-11-04

**Soundness:** 2
**Presentation:** 2
**Contribution:** 1
**Rating:** 3
**Confidence:** 4

**Summary:**

This paper attempts to address input noise in binary classification tasks. The authors provides theoretical analysis of bias due to noisy inputs, and introduce a data augmentation-based correction method to mitigate noise effects. The effectiveness of the proposed method is demonstrated on synthetic and medical image data.

**Strengths:**

The theoretical analysis of the bias introduced by ignoring input noise provides some valuable insights.

**Weaknesses:**

1. Theorem 2 only gives an upper bound on the additional risk, and thus does not tell when input noise cannot be ignored.
2. The theoretical analysis includes some notational complexity that could be simplified or better explained. Notations like $D_{\phi}$ and $D_n$ are confusing.
3. The proposed correction method simply relies on using extra precise data for training, which lacks novelty and is not widely applicable in many practical scenarios.
4. The experiments are only conducted on one real-world dataset, and the proposed method is not compared to any existing state-of-the-art methods.

**Questions:**

See above.

---

### Note · Authors · 2024-11-13

I have read and agree with the venue's withdrawal policy on behalf of myself and my co-authors.